# Trends in Psychiatric Hospitalization of Children and Adolescents in Spain between 2005 and 2015

**DOI:** 10.3390/jcm8122111

**Published:** 2019-12-02

**Authors:** Carlos Llanes-Álvarez, Jesús M. Andrés-de Llano, Ana I. Álvarez-Navares, M. Teresa Pastor-Hidalgo, Carlos Roncero, Manuel A. Franco-Martín

**Affiliations:** 1Department of Psychiatry, Complejo Asistencial de Zamora, 49022 Zamora, Spain; mfrancom@saludcastillayleon.es; 2Department of Pediatrics, Complejo Asistencial Universitario de Palencia, 34005 Palencia, Spain; jm.andres.dellano@gmail.com; 3Department of Psychiatry, University of Salamanca Health Care Complex, 37007 Salamanca, Spain; aialvarez@saludcastillayleon.es (A.I.Á.-N.); croncero@saludcastillayleon.es (C.R.); 4Castilla y León Health Authority, Complejo Asistencial de Zamora, 49022 Zamora, Spain; mtpastor@saludcastillayleon.es

**Keywords:** child psychiatry, eating disorders, big data, hospitalization, decision making, organizational

## Abstract

Eating disorders are on top of chronic conditions in children and adolescents, and the most severe cases may require hospitalization. Inpatient psychiatric treatment is one of the most expensive ones and therefore the efforts when treating eating disorders should focus on avoiding and shortening admissions, as well as preventing readmissions. Advances in of eating disorders treatment lie in an accurate knowledge of those patients requiring admission. This study examined the Conjunto Mínimo Básico de Datos—the largest public hospitalization database in Spain—to estimate the prevalence of eating and other psychiatric disorders during childhood and adolescence. It is a cross-sectional study of the hospital discharges in Castilla y León (Spain) from 2005 to 2015, in which patients under 18 years old with a psychiatric diagnosis at discharge were selected. Trends in the rates of hospitalization/1000 hospitalizations per year were studied by joinpoint regression analysis. Conclusions: eating disorders were the only group that presented an upward and continuous trend throughout the study period. This statistically significant increase showed an annual change of 7.8%.

## 1. Introduction

Childhood and adolescence are high-risk stages for the onset of mental disorders [1,2]. Such disorders and associated problems can make it difficult to acquire developmental milestones and successfully transition children and/or adolescents into adulthood [3,4]. Numerous epidemiological studies have investigated the prevalence of mental disorders in childhood and adolescence, showing that they are relatively frequent among children and adolescents. Their prevalence from childhood to 18 years old is around 25%–30% [5,6].

Hospital care in children and adolescents has proven to be effective [7,8] and continues to be necessary for the comprehensive assistance of children and adolescents. Even so, there are few studies that analyze hospitalization due to psychiatric causes among patients under 18 years of age. The study of trends in psychiatric hospitalization in children and adolescents has not been a priority because these patients represent only around 7% of the total population of hospitalized psychiatric patients [9]. Inpatient psychiatric treatment is one of the most expensive ones, and it is estimated that it represents almost half the annual cost of mental health treatment for children and adolescents [10]. However, and despite the efforts to develop outpatient alternatives, there are cases in which hospitalization is still necessary [11].

Eating disorders have become one of the most common chronic diseases in the pediatric age group, especially among females, and they almost exclusively afflict young persons. The peak ages for these diseases are late adolescence and young adulthood, in both cases under 18 years old. The incidence of eating disorders appears stable overall but may be increasing in younger age groups. Previous studies have demonstrated that the representativeness of the available eating disorder prevalence data is poor [12,13]. In Spain, data on incidence and clinical features of eating disorders are sparse too.

For epidemiological studies on eating disorders there are some methodological issues. Eating disorders are relatively rare among the general population and patients tend to deny or conceal their illnesses and avoid professional help [14], making community studies costly and ineffective. Therefore, many epidemiological studies use psychiatric case registers or medical records from hospitals in a circumscribed area as we have done in this research. This type of study will underestimate the occurrence of eating disorders in the general population, because not all patients will be detected, but will include the most serious cases that are the ones with higher morbidity and mortality in addition to those that will consume more health resources [15]. A large part of the advances in eating disorders treatment have to be focused on this group, during and after hospitalization, so it is necessary to know their characteristics well.

The Conjunto Mínimo Básico de Datos (translated as “basic minimum data set” and hereinafter referred to as CMBD), is the broadest administrative clinical database in Spain. Its completion is mandatory in the public hospitals of the Autonomous Communities that make up our National Health System [16]. Its sample size is huge, and we can say that its analysis would fit into the investigation of the so-called big data. The main advantage is its large sample size, which gives it a very adequate statistical power [17]. The main objective of this study is the statistical exploitation of the Minimum Basic Data Set of Castilla y León to provide epidemiological and clinical information (average age, gender, rural or urban background) and reference parameters on the casuistry and operation of hospitals (diagnosis and average stay), and to compare the trends of hospitalization due to eating disorders with the rest of mental disorders. With 94,223 km^2^, Castilla y León is the largest region in Europe. It has a population of around 2.5 million (2,410,819 inhabitants in 2018) [18], which are distributed in a balanced way between rural and urban areas, and within the latter in large, medium and small cities. It also presents a territorially unbalanced productive structure and economic development. This diversity places Castilla y Leon in a unique context in Spain and Europe for conducting epidemiological studies.

The study of trends in incidence plays a central role in epidemiology and public health [19]. However, only few studies analyze the rates of psychiatric hospitalization during childhood and adolescence; they may help interpret the effectiveness of actual treatments against these disorders and be a model of evaluation of further initiatives. On the other hand, administrative databases, such as hospital admissions have proved useful in obtaining epidemiological information of different processes, in the absence of specific records [20]. The objective is to know, as a method of evaluating the impact of preventive and therapeutic interventions in health, if there are changes in trends in hospitalization rates for eating disorders.

## 2. Experimental Section

We conducted a cross-sectional study of the hospital discharge database of Castilla y León from 2005 to 2015, selecting patients under 18 years old with a psychiatric diagnosis at discharge. Trends in the rates of hospitalization/1000 hospitalizations per year were studied by joinpoint regression analysis.

Sample: the data come from the CMBD and contain very valuable information to know the health reality of a population, since in addition to collecting the usual demographic data (age, sex, urban or rural residence), the CMBD records the diagnosis that has motivated the admission (main diagnosis). Finally, the CMBD includes the patient’s date of admission and discharge, as well as their circumstance of admission (urgent, scheduled) and circumstance of discharge (discharge to his home, death, transfer to another hospital, etc.) [16]. The coding in the CMBD is done based on hospital discharge reports and, since they correlate closely with hospital admission, we will use both terms as synonyms in our work. The target population in the middle of the study period (2010) is made up of 370,762 children under 18 out of a total population of 2,547,408 people. For the standardization by age, the European standard population of 2013 was used.

The study population is made up of 1551 cases of hospital discharges with a main diagnosis of mental disorder, in children and adolescents (under 18), in the public hospitals of Sanidad de Castillay León (SACYL) and between 2005 and 2015, both inclusive. They were classified based on the criteria of the CMBD base of the hospital and the ICD-9-MC.

Variables analyzed: study of hospital discharges from hospitals in Castilla y León between 2005 and 2015. Patients with a primary diagnosis were selected upon discharge from abuse or dependence on psychoactive substances. Main diagnoses for hospital discharges according to selection of ICD-9-MC codes were used in previous research [21,22]. Codes used are detailed in Table 1.

Statistical analysis: General descriptive data for the whole group and for each disease studied. Incidence rates are calculated per 1000 hospitalizations per year, global and specific, by type of substance and the trend over the 11 years studied, in general and by substance groups. The analysis of trends to determine if there were changes in the rates with significant statistical differences over time was performed by linear joinpoint regression, a test that assesses the trend over time in years for the series of selected patients. In this analysis, the points of change (joinpoints or inflection points) show statistically significant changes in the trend (ascending or descending). Graphically, the joinpoint models performed on the logarithm of the rate describe a sequence of connected segments. The point at which these segments come together is a joinpoint and represents a statistically significant change in the trend. In addition, for each segment, an annual change percentage was calculated for each trend by means of generalized linear models, assuming a Poisson distribution and showing in each case its level of associated statistical significance, with 95% confidence intervals (95% CI), and hospitalization rates stratified by gender with their respective 95% CI and their statistical significance.

Software: we used free access software from the Research and Surveillance Program of the National Cancer Institute of the United States [23,24]. Values of *p* < 0.05 were considered statistically significant differences. The statistical analysis was carried out with the SPSS v21.0 program.

The data that support the findings of this research are available at the Dirección General de Sistemas de Información, Calidad y Prestación Farmacéutica, located at: Pseo. de Zorrilla, 1. C.P.: 47007 Valladolid (Spain). Legal restrictions apply to access this data (Ley 16/2003, de 28 de mayo, de cohesión y calidad del Sistema Nacional de Salud), which were used with the relevant authorization in this study. We provide a link to the access conditions https://www.boe.es/eli/es/l/2003/05/28/16.

## 3. Results

The hospital network of Castilla y León comprises 14 centers: 3 regional, 6 provincial and 5 reference centers, structured on their health area and the availability of different medical specialties. 

The CMBD of hospital discharges from Castilla y León, between 2001 and 2015, consists of 3,359,572 records, of which 340,443 were under 18 years old. A total of 52,692 were hospitalizations in psychiatry units (of any age). Finally, 1551 of these correspond to patients under 18 admitted to the psychiatric units of the public centers in Castilla y León. The diseases were selected according to the indicated codes and between 2005 and 2015. The main diagnosis was eating disorders with 371 of 1551 hospitalizations followed by psychosis (257) and behavioral disorders (163). A total of 52,692 hospitalizations for any psychiatric diagnosis were used for calculating the rates (Table 2 and Table 3).

In this sample, 49.8% were male versus 50.2% female. This proportion with respect to sex is similar in the rest of diagnosis groups except for psychosis, hyperactivity and bipolar disorders, which were predominant in men (77.8%, 76.4% and 67.9% respectively). In the case of eating disorders, the percentage of women 81.9% was the highest of all groups.

The mean age of the total sample was 14.4 ± 2.6 years, with a range between 16 ± 2.5 years of average age of the substance abuse group and 12.6 ± 3 years of average age of hyperactivity group. The urban or rural origin remains fairly stable for all groups in an approximate ratio of 7:3 except for the bipolar disorder that approaches parity (58.2% vs. 41.8%) (Table 4).

### Analysis of Hospitalization Rate Trends


Bipolar Disorders Trend Changed in 2007, with Two Upward Trends in 2005–2007 and 2007–2015.Hyperactivity disorders showed two upward trends in 2005–2007 and 2007–2015 with a joinpoint in 2007.Other groups
(1)For psychosis, depressive disorders, adaptive disorders, substance abuse and other disorders, slightly ascending—although statistically non-significant—trends were observed.(2)For anxiety disorders and behavioral disorders groups, a slight downward trend was observed although without statistical significance.Eating disorders was the only group that presented an upward and continuous trend throughout the study period. This statistically significant increase showed an annual percentage of change of 7.8% (Figure 1).


## 4. Discussion

This study provides three novel aspects scarcely published in our area. First of all, there is the use of a database, such as the CMBD for hospital discharges, whose analysis transforms data into useful information for health decision-making both in the context of the years analyzed and in the present, as very little similar research has been published so far [25]. Secondly, this this type of cross-association study is common in epidemiological research and is more than a mere description, within a context of clinical reality such as the discharge of a hospital network. And finally, the statistical methodology used, using joinpoint regression models, is very effective to identify trends and changes in different pathologies over time. The results of the research showed that hospitalization rates remained stable in the 11 years of research for most of the pathologies. The average stay is similar to that reported by other studies although with notable differences between diagnostic groups, with eating disorders [9].

Another remarkable fact is the continuous increasing trend that occurs along 11 years of research with an annual increase of 7.8% for eating disorders. They had the greatest impact on resource consumption with the highest number of discharges (371) and one of the highest average stays (29.2 ± 21.8) (Table 5), according to previous evidence [26]. Our hypotheses are that we face a kind of second wave of improvement in the psychiatric assistance of children and adolescents after a long period in which the main classic psychiatric disorders such as schizophrenia, depression or bipolar disorder have been prioritized, that supply has generated demand or, most likely, a primary increase in the prevalence of eating disorders.

The high proportion of hospitalizations (85.2% between 13–18 years old) that take place in the age group near the age of majority (18 in Spain) reveals the importance of maintaining care continuity [27] (e.g., with specialized units or programs) during the transfer to adult mental health care, as many psychiatry departments have separate teams of child and adult psychiatry [28].

Health managers have tended to traditional approaches, accustomed to functioning independently and relying on their own clinical judgement, sometimes by imitation or by previous operation, and seldom depending on protocols based on big data. This, together with an under-investment in information technology, results in the use of older information systems, with limited ability in standardizing and consolidating data. The current legal restrictions for privacy concerns do not make data sharing easy either. The Spanish healthcare system face unprecedented challenges. Cost pressures have driven health managers to embrace evidence-based medicine. The treatment of ED should be based on a multimodal model [29], which should require knowledge of ED epidemiology, and the planning of ED services should adapt to the characteristics of the local catchment area [30] in order to ensure this heterogeneous clinical population the most appropriate treatment options, rooted in their real clinical history and course.

Limitations: As limitations of the study, it could be considered that the data were obtained retrospectively from a non-specifically clinical administrative record, whose coding could have undergone changes over the years and in the different hospitals. Despite this, the study of databases such as the CMBD, with a large volume of information, is a recognized approach to the knowledge of the reality of a pathology. Dieting behaviors and body image concerns are common in adolescence and it can be challenging to identify those patients at the extreme end of this spectrum who develop an eating disorder. That is why they may be underdiagnosed and hospitalization samples like this guarantee established diagnoses and in cases of severity.

## 5. Conclusions

A total of 53,748 hospital discharges due to psychiatric diseases took place during the research period, and 1551 of those corresponded to patients under 18; 2.9% of the total. Hospitalizations due to eating disorders are the most frequent and long, growing by 7.8% at least until 2015. The innovative statistical methodology of joinpoint regression models may be very useful in identifying trends and changes in different pathologies over time, reducing costs and improving patient outcomes. 

## Figures and Tables

**Figure 1 jcm-08-02111-f001:**
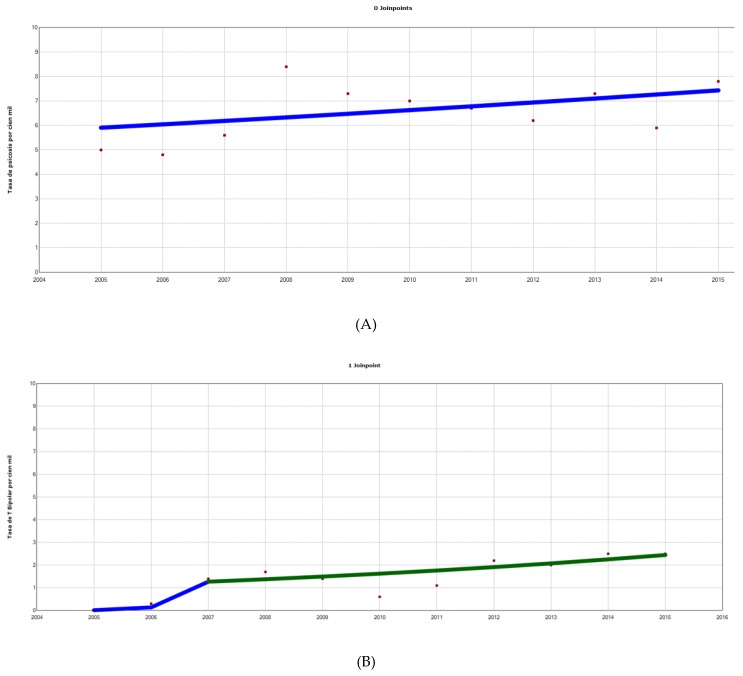
Hospitalization rates per 100,000 inhabitants. Analysis by groups of the diseases studied (**A**–**J**, from left to right and from top to bottom) with inflection points (joinpoints) and APC. (**A**) Psychosis: 0 joinpoints, APC 2005–2015 2.3 (95% CI −1.4 to 6.2, *p* < 0.05 *). (**B**) Bipolar Disorder, 1 joinpoint (2007) APC 2005–2007 827 (95% CI −32.4 to 12,591, *p* = 0.1), APC 2007–2015 8.5 (95% CI −2.7 to 21.1, *p* = 0.1). (**C**) Depressive disorders, 0 joinpoints, APC 2005–2015 2.7 (95% CI –3.3 to 9.1, *p* = 0.3). (**D**) Anxiety disorders, 0 joinpoints, APC 2005–2015 −0.7 (95% CI −9.2 to 8.7, *p* = 0.9). (**E**) Adaptative disorders, 0 joinpoints, APC 2005–2015 4.1 (95% CI −3.8 to 12.5, *p* = 0.3). (**F**) Behavioral disorders, 0 joinpoints, 2005–2015 −0.5 (95% CI −8.4 to 8.1, *p* = 0.9). (**G**) Substance use disorders, 0 joinpoints, APC 2005–2015 6.8 (95% CI −3.4 to 18, *p* = 0.2). (**H**) Hyperactivity disorders, 1 joinpoint (2007) APC 2005–2007 215 (95% CI −56.8 to 2200, *p* = 0.2), APC 2007–2015 0.6 (95% CI −6.0 to 7.7, *p* = 0.8), (**I**) others, 0 joinpoints, APC 2005–2015 2.8 (95% CI −4.4 to10.5). (**J**) Eating disorders: 0 joinpoints, APC 2005–2015 7.8 (95% CI 3.8 to 12, *p* < 0.05 *). APC: annual percentage change; 95% CI: 95% confidence interval. (*): APC statistically significant. Red dots: exact annual value. Lines represent trends, with line colors changing where joinpoints were identified. A blue line only represents a monotonic trend. X-axis: years (from 2004 to 2016). Y-axis: discharge rate for each drug studied; discharge rates for drugs/1000 hospital discharges. APC: annual percentage of change. 95% CI: 95% confidence interval. Data represent exact annual value. * Statistically significant CAP.

**Table 1 jcm-08-02111-t001:** Main diagnoses for hospital discharges and associated ICD-9-MC codes.

No.	Diagnosis Group	ICD-9
I	Psychosis (including depressive psychosis)	295.00–295.99, 297.00–297.99, 298.00–298.99, 299.00–299.99
II	Bipolar disorders	296.00–296.19, 296.40–296.81, 296.89–296.99
III	Depressive (non-psychotic) disorders:	296.20–296.39, 296.82, 300.40–300.59, 301.10, 309.00–309.19, 311.00–311.99
IV	Anxiety disorders	300.00–300.39, 307.20–307.23, 308.00–308.99, 313.00–313.29
V	Adaptation disorders	309.20–309.99
VI	Behavioral disorders	312.00–312.99
VII	Substance abuse	291.00–292.99, 303.00–305.99
VIII	Eating disorders	307.10–307.19, 307.50–307.59
IX	Hyperactivity disorder	314.00–314.99
X	Others	293.00–294.99, 300.60–301.09, 301.11–302.99, 306.00–307.09, 307.40–307.00, 307.60–307.99, 313.30–313.99, 315.00–319.99

**Table 2 jcm-08-02111-t002:** Annual population distribution, number of total hospital discharges by psychiatric cause as well as hospital discharge rate per 100,000 inhabitants and year adjusted by age.

Year	Population of Castilla y León (under 18 Years Old)	Total Child Discharges (n)	Psychiatric Discharges (under 18 Years Old)	Annual Rate (Discharges by Psychiatric Pathology in Children under 18/100,000)
2005	361,237	23,208	99	27.7
2006	361,145	24,015	92	25.8
2007	363,298	24,517	136	38.1
2008	367,478	23,083	170	45.6
2009	370,396	22,549	140	39.2
2010	370,762	22,583	132	36.9
2011	370,362	22,381	135	4.8
2012	369,460	21,016	154	43.2
2013	367,697	21,443	145	40.7
2014	364,334	20,884	170	47.6
2015	358,788	20,842	178	49.8
	TOTAL	340,443	1551	

**Table 3 jcm-08-02111-t003:** Hospitalization rates for the different processes studied per 100,000 inhabitants and year adjusted for age.

Year	Psychosis	Bipolar Disorder	Depressive Disorders	Anxiety Disorders	Adaptive Disorders	Behavioral Disorders	Substance Abuse	Eating Disorders	Hyperactive Disorders	Others
2005	5	0	2.2	3	3.6	2.5	0.8	7.6	0.3	2.5
2006	4.8	0.3	2.5	1.4	2	4.2	1.1	5.6	1.7	2.2
2007	5.6	1.4	2	2.5	3.4	5.6	1.4	8.2	4.7	3.3
2008	8.4	1.7	2.5	5.6	2.8	6.7	0.3	9	5.6	5
2009	7.3	1.4	1.1	4.8	3.9	3.9	1.7	5.6	4.5	5
2010	7	0.6	1.4	2.2	5.9	3.9	1.1	9	3.4	2.5
2011	6.7	1.1	2	2	3.9	3.1	1.7	7.9	4.7	4.8
2012	6.2	2.2	2	3.4	3.9	5	1.4	13.2	4	2
2013	7.3	2	2.5	2.2	3.9	2.5	0.8	12.1	4.2	3.1
2014	5.9	2.5	3.7	3.4	7.3	2.2	0.3	12.6	5.9	3.9
2015	7.8	2.5	2.2	3.1	2.5	5.9	2.5	13.5	5	4.8

**Table 4 jcm-08-02111-t004:** Characteristics of the cases analyzed, such as psychiatric pathology in general and for each of the processes studied.

	Psychosis	Bipolar Disorder	Depressive Disorders	Anxiety Disorders	Adaptive Disorders	Behavioral Disorders	Substance Abuse	Eating Disorders	Hyperactive Disorders	Others	Total
Cases (n)	257	56	86	120	154	163	47	371	157	140	1551
Average age years (SD)	14.6 (2.6)	15.1 (2.4)	15.3 (1.9)	14.1 (3.1)	14.4 (2.7)	14.7 (2.0)	16 (2.5)	14.6 (2.1)	12.6 (3.0)	14.3 (3.0)	14.4 (2.6)
**Age (%)**											
0–6	1.2%	0%	0%	2.5%	1.3%	0.6%	2.1%	0.3%	1.9%	2.1%	1.1%
7–12	15.6%	12.5%	9.3%	23.3%	16.9%	8.0%	0%	14.6%	41.4%	19.3%	17.3%
13–18	83.3%	87.5%	90.7%	74.2%	81.8%	91.4%	97.9%	85.2%	56.7%	78.6%	81.6%
**Sex (%)**											
Men	77.8%	67.9%	36%	49.2%	40.9%	54%	66%	18.1%	76.4%	53.6%	49.8%
Women	22.2%	32.1%	64%	50.8%	50.1%	46%	34%	81.9%	23.6%	46.4%	50.2%
**Area**											
Urban	79.6%	58.2%	70.9%	67.8%	77.3%	66.3%	85.1%	75.6%	72.3%	73.9%	73.8%
Rural	20.4%	41.8%	29.1%	32.2%	22.7%	33.7%	14.9%	24.4%	27.7%	26.1%	26.2%

**Table 5 jcm-08-02111-t005:** Average stay in days per process studied and total number of days of accumulated stay per process.

	Psychosis	Bipolar Disorder	Depressive Disorders	Anxiety Disorders	Adaptive Disorders	Behavioral Disorders	Substance Abuse	Eating Disorders	Hyperactive Disorders	Others	Total
Average stay days (DE)	19.6 (28.5)	19.8 (12.3)	14.5 (13.9)	14.7 (13.5)	12.6 (9.0)	8.9 (9.2)	11.2 (8.2)	29.2 (21.8)	18.4 (15.5)	13.8 (18.2)	18.5 (19.9)

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
