# Peer review of "Trends in Psychiatric Hospitalization of Children and Adolescents in Spain between 2005 and 2015"

_jcm, 2019, doi:10.3390/jcm8122111_

Round 1

Reviewer 1 Report

It was good to be able to read a good paper faster than others while reviewing it.

This is a very interesting paper, because in particular, there have been data on eating disorders in youth from outpatient

clinic, but inpatients data are rare. However, there is  no explanation at all why the rate of admission for eating disorders is increasing. It would be a much better study if the authors could give readers hypotheses or the reasons about whether the prevalence of youth with eating disorders has actually increased, or whether it is due to changes in the local health care system, and so on.

And unfortunately, what's sad about the paper review is that figure 1 is a very important result, but the resolution is low,

therefore, I can not check it properly.

Author Response

REVIEWER 1

Dear reviewer,

We would like to thank the reviewer for careful and thorough reading of this manuscript and for the thoughtful comments and constructive suggestions, which help to improve the quality of this manuscript. Our response follows (your comments are in italics).

C1 It was good to be able to read a good paper faster than others while reviewing it.

We appreciate the positive feedback from the reviewer. With regards to the length of the article it was our main goal to write a short article but that contains so much information. We tried to target both readers; those who are interested in having a global vision of the current state of psychiatric hospitalization in child and adolescents and readers that are only interested in subsections (psychosis, bipolar disorders…, or as in this case eating disorders). The current form serves both of these needs and with the changes in the figures that we have introduced in this second round will be specially easy to read to the research who is only interested in one or a few sections. Thank you for appreciating our efforts.

You will find that the manuscript has two additional pages than the previous one due to the readjustment of the figures.

C2 This is a very interesting paper, because in particular, there have been data on eating disorders in youth from outpatient, but inpatients data are rare.

After a thorough literature review we have barely found comparable studies. Psychiatric hospitalizations of children and adolescents are rare, and long multicentre studies are required to have considerable sample like this.

C3 However, there is no explanation at all why the rate of admission for eating disorders is increasing. It would be a much better study if the authors could give readers hypotheses or the reasons about whether the prevalence of youth with eating disorders has actually increased, or whether it is due to changes in the local health care system, and so on.

Thank you for this suggestion; indeed it would have been interesting to explore why this increase occurs and we have incorporated your suggestion throughout the manuscript. We don't know for sure what this marked increase responds to but our main hypotheses are:

- There is a primary increase in the prevalence of eating disorders as the previous literature seems to indicate: Litmanen J, Fröjd S, Marttunen M, Isomaa R, Kaltiala-Heino R. Are eating disorders and their symptoms increasing in prevalence among adolescent population? Nord J Psychiatry. 2017 Jan;71(1):61–6.

- In the past, efforts have been focused on improving the assistance of children and adolescents affected by the main classic psychiatric disorders such as schizophrenia, depression or bipolar disorder but not by eating disorders. In recent years, eating disorders have been incorporated into this “second wave” of improvement in the Psychiatric assistance of children and adolescents, what has led to a relative increase in hospitalizations for these disorders.

-In clinical practice, hospitalization is not based only on clinical criteria; other factors are involved. The creation of psychiatric hospitalization units for children and adolescents has made available to clinicians a therapeutic resource that they previously lacked, and so on, supply has generated demand. Of course there will always be interferences in the results due to the idiosyncrasy of the region and the local health system but, a growing and sustained trend over 11 years, that is not observed in other disorders, has to have a clinical meaning.

Although they are only hypotheses we have added to the discussion a brief description of them.

C4 And unfortunately, what's sad about the paper review is that figure 1 is a very important result, but the resolution is low, therefore, I can not check it properly.

We would like to apologize for the inconvenience this mistake with the resolution of the figures may have cause, and we thank the reviewer for pointing that out.

Graphics have been a subject of discussion in our team. We conceive this article as a research of epidemiology in eating (and other) disorders in children and adolescents, and although we have tried to make it as enjoyable as possible, it is just what it is. On the one hand the figures may confuse or get the reader bored, but on the other hand they are the most efficient way to condense such amount of information. We hope you find that the manuscript don`t contain too many figures and that this might distract the reader from the main messages. We have readjusted the graphics that are now presented in a single column instead of two as in the previous version. It’s harder now to take a quick look and get a comprehensive overall picture of the trends of these disorders, but the resolution is much better.

In addition to the above comments, all spelling and grammatical errors pointed out by the reviewers have been corrected. We regret if there were problems with the English. As there isn’t a native English-speaking co-author to thoroughly revise the grammar of this manuscript, The paper has been carefully revised by Mr. F. Bautista-Becerro, (Ph.D. in Pharmacy and Translation and Interpretation) who provides a professional language editing service, and Mr J. Bradley (a native English speaker) to improve the grammar and readability.

We look forward to hearing from you in due time regarding our submission and we hope that you find our responses satisfactory and that the manuscript is now acceptable for publication, or respond to any further questions and comments you may have.

Sincerely,

The authors

Reviewer 2 Report

The objective of this paper "... is to know, as a method of evaluating the impact in health preventive and therapeutic interventions if there changes in trends in hospitalization rates hospitalization for eating disorders." Which I interpreted to mean that by analyzing the data provided by the Conjunto Minimo Basico de Datos (CMBD), the authors would be looking at trends in the hospitalization rates for children and adolescents diagnosed with eating disorders, as compared to other mental disorders,  in order to establish the impact of the preventive and therapeutic health interventions.

The strengths of the paper are (1) access to this unique data set, albeit administrative, still very informative for epidemiological purposes; (2) the population studied is adequately described and inferences can be drawn to other populations because of the heterogeneity of the population; (3) the use of innovative statistical method that can establish trends and therefore be used to allocate resources in a knowledgeable way to the areas where it is needed.

On the down side, it is a paper that is difficult to read because English is not the author's first language, some of the details are lost in translation and take away from the readers ability to appreciate the methodology and implications. I would also recommend being consistent across the paper in how things are labeled (ie. using the term "disorder" throughout as opposed to process, disease, etc.) and to avoid colloquialisms that might not apply or be understood by a wider readership for example describing the age of the population using the "age of majority". The other big issue noted is that the figures are illegible...

    In the age of big data, knowing how to interpret it and use it to our advantage is key, this paper tries to make a case for using administrative data within a clinical context to better allocate resources to effectively change the trends and the best practices when dealing, in this case, with eating disorders.

Author Response

REVIEWER 2

We would like to thank the reviewer for careful and thorough reading of this manuscript and for the thoughtful comments and constructive suggestions, which help to improve the quality of this manuscript. Our response follows (your comments are in italics).

C1 The objective of this paper "... is to know, as a method of evaluating the impact in health preventive and therapeutic interventions if there changes in trends in hospitalization rates hospitalization for eating disorders." Which I interpreted to mean that by analyzing the data provided by the Conjunto Minimo Basico de Datos (CMBD), the authors would be looking at trends in the hospitalization rates for children and adolescents diagnosed with eating disorders, as compared to other mental disorders,  in order to establish the impact of the preventive and therapeutic health interventions.

We appreciate the positive feedback from the reviewer. We tried to target both readers; those who are interested in having a global vision of the current state of psychiatric hospitalization in child and adolescents and readers that are only interested in subsections (psychosis, bipolar disorders…, or as in this case eating disorders). The current form serves both of these needs and with the changes in the figures that we have introduced in this second round will be specially easy to read to the research who is only interested in one or a few sections. Thank you for appreciating our efforts.

With regards to the length of the article it was our main goal to write a short article but that contains so much information. You will find however that the manuscript has two additional pages than the previous one due to the readjustment of the figures.

C2 The strengths of the paper are (1) access to this unique data set, albeit administrative, still very informative for epidemiological purposes; (2) the population studied is adequately described and inferences can be drawn to other populations because of the heterogeneity of the population; (3) the use of innovative statistical method that can establish trends and therefore be used to allocate resources in a knowledgeable way to the areas where it is needed.

After a thorough literature review we have barely found comparable studies. Psychiatric hospitalizations of children and adolescents are rare, and long multicentre studies are required to have a considerable sample like this.

But the merit it is not just ours, the health authorities of our region have trusted us by depositing such an extensive and sensitive database. For this reason, we have had the responsibility of guarding and making careful use of this information but also of analyzing it correctly; this analysis may lead future health policies and interventions, resources that may finance one intervention or another.  But more difficult than analyzing the data is to interpret the results correctly. It would be interesting to explore why this growth occurs and we have incorporated some ideas trying to enrich the manuscript. We don't know for sure what this marked increase responds to, but of our knowledge of the health system and from the experience of some of us as heads of department and health managers the main hypotheses are:

- There is a primary increase in the prevalence of eating disorders as the previous literature seems to indicate: Litmanen J, Fröjd S, Marttunen M, Isomaa R, Kaltiala-Heino R. Are eating disorders and their symptoms increasing in prevalence among adolescent population? Nord J Psychiatry. 2017 Jan;71(1):61–6.

- In the past, efforts have been focused on improving the assistance of children and adolescents affected by the main classic psychiatric disorders such as schizophrenia, depression or bipolar disorder but not by eating disorders. In recent years, eating disorders have been incorporated into this “second wave” of improvement in the Psychiatric assistance of children and adolescents, what has led to a relative increase in hospitalizations for these disorders.

-In clinical practice, hospitalization is not based only on clinical criteria; other factors are involved. The creation of psychiatric hospitalization units for children and adolescents has made available to clinicians a therapeutic resource that they previously lacked, and so on, supply has generated demand. Of course there will always be interferences in the results due to the idiosyncrasy of the region and the local health system but, a growing and sustained trend over 11 years, that is not observed in other disorders, has to have a clinical meaning.

Although they are only hypotheses we have added to the discussion a brief description of them

C3 On the down side, it is a paper that is difficult to read because English is not the author's first language, some of the details are lost in translation and take away from the readers ability to appreciate the methodology and implications. I would also recommend being consistent across the paper in how things are labeled (ie. using the term "disorder" throughout as opposed to process, disease, etc.) and to avoid colloquialisms that might not apply or be understood by a wider readership for example describing the age of the population using the "age of majority". The other big issue noted is that the figures are illegible...

In addition to the above comments, all spelling and grammatical errors pointed out by the reviewers have been corrected. We regret if there were problems with the English. As there isn’t a native English-speaking co-author to thoroughly revise the grammar of this manuscript, the paper has been carefully revised by Mr. F. Bautista-Becerro, (Ph.D. in Pharmacy and Translation and Interpretation) who provides a professional language editing service, and Mr J. Bradley (a native English speaker) to improve the grammar and readability. We have also tried to improve  the coherence across the paper in how things are labeled.

We would like to apologize for the inconvenience this mistake with the resolution of the figures may have cause, and we thank the reviewer for pointing that out.

Graphics have been a subject of discussion in our team. We conceive this article as a research of epidemiology in eating (and other) disorders in children and adolescents, and although we have tried to make it as enjoyable as possible, it is just what it is. On the one hand the figures may confuse or get the reader bored, but on the other hand they are the most efficient way to condense such amount of information. We hope you find that the manuscript don`t contain too many figures and that this might distract the reader from the main messages. We have readjusted the graphics that are now presented in a single column instead of two as in the previous version. It’s harder now to take a quick look and get a comprehensive overall picture of the trends of these disorders, but the resolution is much better.

C4 In the age of big data, knowing how to interpret it and use it to our advantage is key, this paper tries to make a case for using administrative data within a clinical context to better allocate resources to effectively change the trends and the best practices when dealing, in this case, with eating disorders.

That is indeed the essence of work thank you for taking the time and effort necessary to review, We look forward to hearing from you in due time regarding our submission and we hope that you find our responses satisfactory and that the manuscript is now acceptable for publication, or respond to any further questions and comments you may have.

Sincerely,

The authors
